# "We couldn't think in the box if we tried. We can't even find the damn box": A qualitative study of the lived experiences of autistic adults and relatives of autistic adults

**Tracy L. Finch**[1]*, **Joan Mackintosh**[2], **Alex Petrou**[3], **Helen McConachie**[3], **Ann Le Couteur**[3], **Deborah Garland**[4], **Jeremy R. Parr**[3,5]

1 Department of Nursing, Midwifery & Health, Northumbria University, Newcastle upon Tyne, United Kingdom, 2 Newcastle University, Population Health Sciences Institute, Newcastle University, Newcastle Upon Tyne, United Kingdom, 3 Newcastle University, Population Health Sciences Institute, Newcastle University, Sir James Spence Institute, Royal Victoria Infirmary, Newcastle Upon Tyne, United Kingdom, 4 National Autistic Society, London, United Kingdom, 5 Cumbria, Northumberland Tyne and Wear NHS Foundation Trust, Newcastle upon Tyne, United Kingdom

* tracy.finch@northumbria.ac.uk

**Data Availability Statement:** Data cannot be shared publicly as informed consent from participants was not obtained for this purpose. Due

## Abstract

Autistic children grow to become autistic adults, and autism is increasingly diagnosed in adulthood and later life. This qualitative study aimed to understand experiences of autism throughout adulthood. A national cohort study of autistic adults and relatives of autistic adults (ASC-UK), enabled purposive recruitment of a diverse sample. Semi-structured interviews were conducted with 29 autistic adults (aged 20–71 years), mostly diagnosed in adulthood, and 16 relatives (aged 31–81 years) of autistic adults diagnosed across both childhood and adulthood (including some with learning disability). Interview topics included health, relationships, education, employment, quality of life and everyday experiences. Thematic analysis of the accounts of the autistic adults identified six key themes relating to their experiences: (1) diagnosis as validating yet limiting; (2) supportive and non-supportive social agents; (3) the "invisibility" of the needs of autistic adults; (4) health in the context of autism; (5) staying 'outside' the circle; and (6) multiple lives with autism. Data from relatives about autistic adult experiences gave additional perspectives on these themes. Experiences reported in other studies–of 'difference' from others, challenges of social engagement, and learning to 'conform' to society's expectations–were evident and relevant to male and female autistic adults, across all age groups, and unrelated to stage of life when diagnosed. Some expressed disappointment with their lives, but others were proud of their achievements. Education and employment, whilst challenging for many, were also rewarding for some. Health care and social services were often experienced as inaccessible, inappropriate, or lacking understanding of the individual's needs. We conclude that greater public understanding of autism as experienced in adulthood is needed. Key priorities are improving the availability of 'appropriate' health and social care services for autistic adults and families, and providing practical support to enable enhanced participation in life.

to the highly personal and sensitive nature of participants' accounts in these interviews, the removal of all contextual data as necessary to protect participants' identities makes the transcripts unintelligible. The data on which this paper reports cannot therefore be made publicly available. Data are available from the Independent Chairperson of the Autism, Adulthood and Ageing Steering Committee (contact via adultautismspectrum@newcastle.ac.uk) for researchers who meet the criteria for access to confidential data.

**Funding:** The ASC-UK study (awarded to JP, HMcC, AL & TF) is funded by Autistica:https://www.autistica.org.uk/our-research/research-projects/investigating-autistic-adults-across-their-lives-and-into-old-age The funders had no role in study design, data collection and analysis, decision to publish, or preparation of the manuscript.

**Competing interests:** The authors have declared that no competing interests exist.

## Introduction

It is estimated that around 1% of the UK population has autism [1,2]. Autism is a lifelong neurodevelopmental condition and while those presenting with more severe characteristics are likely to have had their autism recognised at an early age [3], there is still relatively little known about the lives of adults on the autism spectrum, whether diagnosed in childhood or later life. Understanding better the experiences (including aspirations, achievements and needs) of autistic adults and family members across the lifecourse is a priority for autistic individuals, their families, policy makers, service providers and researchers [4,5].

In recent years, research has shifted to include a focus on autism in adulthood [4,6–10]. Whilst a number of studies have considered adult outcomes and their predictors [5,11,12], large scale empirical research into the life experiences of autistic adults remains limited [13]. Recent qualitative research exploring young autistic adults and their parents' experiences and expectations for transitioning into adulthood [14] suggests that previous conceputalisations of 'outcomes' and the normative standards against which the developmental achievements of autistic indivdiduals can meaningfully be judged, require revision. The contribution of qualitative research for understanding autism and the personal experiences of individuals is becoming better acknowledged [15]. In 2014, Pellicano and colleagues [16] investigated whether existing research in autism reflected priorities and concerns of the community, undertaking interviews and focus groups with autistic adults, family members, practitioners and researchers alongside a survey of stakeholders. They reported that autism research was seen by autistic adults to represent *'neurotypical priorities regarding us'*, rather than addressing research questions that they would identify as important to making differences in their own day to day lives. Survey research conducted by Gotham et al [17] with autistic adults across the lifecourse (aged 18–71 years), identified some key priorities including improving public services and healthcare access. Participants have also called for greater public acceptance of autism [18] and recognition that a more person-centred approach is necessary in order to understand for example how individuals might define success in terms of personal achievements as well as other life experiences such as accessing healthcare and community services [19].

Against this back-drop, whilst qualitative studies of the current contemporaneous life experiences of adults remain limited [10], qualitative studies focusing on particular topics and with particular autistic populations, are beginning to offer key insights on a number of important aspects of the lives of autistic adults. The experiences of autistic persons of 'trying to fit in' (with others and society in general), and of 'wearing a mask' to hide their autistic selves–often referred to as *camouflaging*–are becoming well documented in the literature, and the impacts of this on individuals are beginning to be explored [10,20,21]. Research is also beginning to advance our understanding of the importance of a diagnosis of autism in adulthood, and the impacts that this has–both positive and negative—on invididuals in terms of helping to make sense of themselves and their lives, and on the relationships that they have with others [22,23]. To date most of the qualitative literature exploring the experiences of autism in adulthood has been limited to particular groups of individuals, such as late-diagnosed women [9]; those over the age of 50 [10,22]; participants with Asperger Syndrome [6]; those transitioning into adulthood [8,24]; or on specific topics, for example, experiences of social challenges of autism in adulthood [7] or transitioning into employment [11,25].

Greater involvement of autistic individuals in the design of services is now widely recognised as essential to improve public services and healthcare access, together with greater public acceptance of the needs of autistic people [18]. In the UK, although policy initiatives [26–32] have focused on improving services, current provision of diagnostic, treatment and support services as well as access to employment and community activities is patchy [33]. It is reported

for instance that 80% of adults have difficulty accessing diagnostic services [34], and when physical and mental health services provide 'reasonable adjustments' those that are desirable are infrequently provided [35]. More nuanced person-centred approaches to understanding the aspirations, experiences, achievements and needs of autistic adults are needed to inform and expand the knowledge of the general public including business leaders and employers; as well as funding authorities and providers of post secondary education;community services; health and social care providers; and charitable and third sector organisations.

To contribute to this wider perspective, we undertook a programme of research to better understand the lives of autistic adults; this included establishing a UK adult autism cohort with both quantitative and qualitative research studies, and an investigation of key factors affecting quality of life. In this paper, we report findings from the main qualitative study, that aimed to gain a better understanding of the everyday experiences of autistic adults through first-hand accounts from autistic adults and also from relatives interviewed from the perspective of supporting autistic adults. This study significantly contributes to the existing literature in two key ways. Firstly, we undertook a large and comprehensive study of the experiences of autistic adults spanning a wider age group than has been studied in qualitative research to date. Secondly, we approached the study of autistic adults' experiences without presupposing what aspects of their lives and experiences are most important. Rather we invited participants to explore aspects of their lives that were important to them, from a broad range of topics.

## Methods

### Participants

Participants were recruited from an ongoing longitudinal study of the life experiences of autistic adults, the Adult Autism Spectrum Cohort study (ASC-UK; http://www.autismspectrum-uk.com/). The ASC-UK project based at Newcastle University has recruited more than 4000 participants (over 3000 autistic adults and more than 800 relatives of autistic adults by August 2021) from across the UK from a variety of sources (National Health Service (NHS) diagnostic and mental health teams, NHS primary care (general practitioner (GP)/family practitioners); voluntary sector organisations; and social media). In the ASC-UK study, relatives of autistic adults were included to facilitate the inclusion in ongoing research, of the perspective of adults on the autism spectrum across the ability range including those who were unable to report about their lives due to intellectual disability or for other reasons, in accordance with the NHS Constitution (Crown copyright 2015). In the qualitative study, relatives were interviewed both about their autistic adult relative (reported in this paper), and about their experiences of supporting an autistic relative (the latter focus not reported here). Since published studies concerning autistic adults most commonly focus on participants in the average level of intellectual ability[36], the inclusion of accounts from relatives of autistic adults provided an opportunity to extend the focus of the qualitative study to include the experiences of an additional sample of autistic adults some of whom were not able to report about their own lives [26]. Whilst this inclusion extends the size and general diversity of sample on whom we report, broader representation of autistic individuals with learning disability is not assumed.

All interviewed participants had previously completed an ASC-UK registration form and a series of self-report questionnaires about themselves or in the case of relatives, about the autistic adult relative. They also consented to be approached about other studies including this qualitative interview study. Thus criteria for participation in this study included: membership of the ASC-UK and willingness to be approached about further research; residency in the UK; aged over 18 years; clinical diagnosis of autism, or being a relative of an adult with a clinical diagnosis of any Autism Spectrum Disorder; able to communicate verbally in English.

## Materials

Potential participants received an invitation letter along with a study information leaflet and a consent form for the qualitative study. The qualitative interviewer used one of two interview topic guides (one for autistic adult or a relatives' version) of open and closed questions previously co-developed and piloted with autistic adults who advised the research team. Interview topic guides are included as Appendices 1 and 2. The topics included experiences of: diagnostic assessment and post diagnostic support; autism and physical and mental health: personal relationships and living arrangements; social communication; education; transition from children to adult services; employment; levels/types of social support; and any additional issues identified by the participant.

## Participant recruitment

Participants were recruited from the ASC-UK cohort in waves (around 5 people at a time) to achieve a maximum variation sample (age range, diagnoses and geographical locations). Austic adults were recruited from across four urban/semi-rural geographical areas: North East England, North West England, Midlands, and South East England. Relatives were recruited predominantly from North East England. Sampling continued until no new themes were emerging from the data. All participants were contacted by telephone, post or email (according to their preference indicated at time of joining ASC-UK) and invited to take part in a one-to-one interview.

## Data collection

Participants were offered an opportunity to ask any questions about the study prior to giving informed written consent. Preferences for type of interview (i.e. face-to-face, telephone, email) and location were agreed in advance. All interviews (lasting 37–91 minutes) were audio-recorded. Interviews were conducted by an experienced qualitative researcher with additional training for working with autistic research participants. Reasonable adjustments were made to accommodate the sensory needs and preferences of each participant. The study received a favourable ethical opinion from the Wales Research Ethics Committee 5 [(14/WA/1066)].

## Analysis

All interviews were fully transcribed, checked and anonymised. Data analysis was conducted jointly with members of the research team [37]. Initially, JM and TF undertook thematic analysis following the Framework method [38], using constant comparison and deviant case analysis to enhance internal validity. Typologies were derived, and descriptive and explanatory categories developed. JM and TF independently coded four (14%) adult transcripts and two (13%) relative transcripts, discussed and compared interpretations of the data, and agreed an initial coding framework–a structure of data codes organised under thematic categories. This coding framework was applied to the remaining transcripts by JM, with adjustments to the framework based on joint discussion of data (JM and TF) as data coding progressed. Credibility checks were undertaken by HMc and JP who independently read two of the adult transcripts and two of the relative transcripts and met with JM and TF for consensus discussions. The same coding and analysis procedure was applied to both datasets (adults and relatives), but a full set of analytical themes was developed separately for each of the groups to represent the entire respective datasets. During the analysis process, key emerging findings were presented and discussed at the cohort study steering group meetings, for interpretative contributions from the wider research team that included several autistic individuals.

The separate analyses were mapped against each other, with particular attention to the themes under 'experiences of the adult'. A set of general themes under this category was common to both datasets. Although description of the constituent themes was more detailed in the adults' dataset, there were no additional themes to contribute to the analysis from the relatives' dataset. Therefore, from the analysis of relatives' interviews, data corresponding to relatives's reports of the experiences of the autistic adults are included within the central themes identified in the adults' dataset. This enabled us to retain the focus on the perspectives from the accounts of autistic adults, but with additional opportunities to consider the reports of a different sample of autistic adult experiences provided by family members who may also provide support and take on advocacy roles for their autistic adult relatives. The experiences of the relatives themselves will be reported elsewhere (Hamilton et al, in preparation).

In this paper, participants are described using a pseudonym initial, followed by age at interview, autism spectrum diagnosis, and age at diagnosis (AaD). Identifiers for relative participants include information about their relationship to the autistic adult about whom they are reporting. Illustrative quotes were chosen by the research team to respresent the themes and these are given in the text.

## Results

Interviews were conducted with 29 autistic adults (16 males, 13 females), and 16 relatives (3 males, 13 females) of autistic adults (13 males, 3 females). Details of individual participants are provided in Tables 1 and 2.

Autistic adults aged 20–71 years (median age 43) were recruited from four geographical regions across England: North East England (n = 18), North West England, (n = 5); Midlands (n = 4), and South East England (n = 2). The weighing of participants in favour of North East England reflects both the profile of the cohort database, and the limited resources available to undertake interviews outside of the study region. Autistic adults had a clinical diagnosis of autism given from age 6–67 years (median 41 years); two adults received their diagnosis in childhood. Autistic adults diagnoses were reported as Asperger Syndrome (n = 17); Autism (n = 11) and Autism Spectrum Disorder (ASD) (n = 1). Three participants were additionally diagnosed with dyslexia (two of whom also reported dyspraxia, and one reported an intellectual disability diagnosis. The majority of adults (n = 22) were interviewed within three years of their diagnosis. Participants described themselves as in employment (including self-employed) (n = 12); unemployed (n = 11); retired (n = 5); or at college (n = 2). Over half were living with a partner or family members (n = 17), but many (n = 12) were living alone/independently. Self-reported clinical diagnosis of co-occurring mental health conditions was common: depression (n = 19); anxiety (n = 15); attention-deficit hyperactivity disorder (ADHD) (n = 6); epilepsy (n = 3); bipolar (n = 5); and obsessive compulsive disorder (OCD) (n = 3). Autistic adult participants reported having achieved the following levels of educational attainment: higher education awards (n = 15); high school qualifications (General Certificate of Secondary Education (GCSE), typically undertaken aged 14–16 years, or Advanced Level (A-Level), typically undertaken aged 16–18 years for entry into further education (n = 5); basic school qualification (n = 5); or no formal school qualification (n = 4). Thirteen interviews with autistic adults took place in participants' own homes, ten on university premises, two in a hospital setting, and three in public settings. One autistic adult provided detailed hand-written responses sent by post to the research team: it was not possible to include these data in the present analysis.

The relatives were aged 31-81years (median age 59 years) and included mothers (n = 10) fathers (n = 3); grandparents (n = 1); spouses (n = 1); and a sibling (n = 1) (see Table 2). These

**Table 1. Participant characteristics–autistic adults.**

| Pseudonym | Age at interview | Age at diagnosis | Autism spectrum diagnosis & LD | Co-occurring mental health diagnoses reported | Education level Attained | Employment status | Living status |
|---|---|---|---|---|---|---|---|
| Mr M | 68 | 67 | AS | Depression, anxiety, Epilepsy | Higher education | Self-employed | Lives with wife |
| Mr S | 43 | 41 | AS | Depression, anxiety | No formal qualification | Unemployed | Lives with partner |
| Mr G | 59 | 52 | AS | Depression, Bipolar | Higher education | Retired on health grounds | Lives with wife and children |
| Ms A | 56 | 42 | AS | Depression, Bipolar, OCD | Higher education | Retired on health grounds | Lives with husband |
| Mr D | 43 | 40 | AS | Depression, Anxiety, OCD | High School Advanced Level | Employed | Lives alone, independently |
| Ms L | 59 | 57 | Autism | Anxiety, Epilepsy | No formal qualification | Unemployed | Lives with husband and children |
| Ms B | 37 | 34 | AS | Depression, Anxiety | Higher education | Unemployed | Lives with parents |
| Mr Q | 71 | 46 | Autism | Bipolar | No formal qualification | Retired | Lives alone independently |
| Mr T | 27 | 25 | AS | | Higher education | Employed | Lives with partner |
| Ms D | 53 | 51 | AS | Depression | Higher education | Employed | Lives alone, independently |
| Mr I | 47 | 46 | AS | | Higher education | Employed | Lives alone, independently |
| Ms R | 39 | 39 | Autism | Anxiety | Higher education | Employed (temp) | Lives with husband |
| Ms N | 43 | 41 | Autism | Depression, Alcoholism | Higher education | Employed (p/t) | Lives with husband and children |
| Ms F | 35 | 27 | AS & ID | Depression, Anxiety | High School Advanced Level | Employed (Agency work) | Lives alone, independently |
| Ms X | 65 | 63 | Autism | Depression | Basic qualification | Retired | Lives alone, independently |
| Ms E | 36 | 33 | Autism | Depression, Anxiety, Epilepsy | Higher education | Unemployed | Lives with parents |
| Mr C | 47 | 41 | AS | Bipolar | High School Advanced Level | Employed | Lives with wife and children |
| Mr W | 61 | 58 | Autism | | Higher education | Unemployed | Lives alone, supported |
| Ms H | 46 | 43 | AS | Depression, Anxiety, ADHD | Higher education | Retired on health grounds | Lives with husband and children |
| Mr O | 29 | 25 | Autism | Depression, Anxiety, ADHD, Insomnia | Basic qualification | College & voluntary work | Lives alone, has Personal Assistant |
| Ms K | 23 | 22 | Autism | Depression, Anxiety | Higher education | Unemployed | Lives with parents |
| Ms V | 54 | 52 | AS | Depression, Anxiety, OCD | Basic qualification | Unemployed | Lives alone, independently |
| Ms Y | 24 | 23 | AS | Depression, Anxiety, ADHD | High School GCSE | Unemployed | Married, has support |
| Mr J | 40 | 38 | Autism, Dyslexia & Dyspraxia | Depression | Basic qualification | Unemployed | Lives alone, independently |
| Mr U | 20 | 6 | Autism, Dyslexia & Dyspraxia | ADHD | Basic qualification | College & part time work | Lives with parents |
| Mr P | 49 | 47 | AS | Depression, Anxiety, OCD | Higher education | Employed | Lives with wife and children |
| Mr Z | 25 | 7 | AS | ADHD | High School Advanced Level | Employed | Lives with parents |
| Mr X | 66 | 64 | ASD & Dyslexia | Depression, Anxiety, ADHD, Stroke | No formal qualification | Unemployed | Lives alone, has PA |

(*Continued*)

**Table 1.** (Continued)

| Pseudonym | Age at interview | Age at diagnosis | Autism spectrum diagnosis & LD | Co-occurring mental health diagnoses reported | Education level Attained | Employment status | Living status |
|---|---|---|---|---|---|---|---|
| Mr A | 48 | 45 | AS | Bipolar | Higher Education | Unemployed | Lives alone, independently |

**Abbreviations:** Asperger Syndrome (AS); Autism Spectrum Disorder (ASD); Obsessive Compulsive Disorder (OCD); Attention Deficit Hyperactivity Disorder (ADHD); Learning Disability (LD); Intellectual Disability (ID).

relatives reported on a separate group of autistic adults diagnosed at age 3–55 years (median age at diagnosis 19 years). These participants were recruited primarily from North East England (n = 14). Of the autistic adults they spoke of, age of autism diagnosis was mixed,

**Table 2. Participant characteristics–relatives of autistic adults and autistic adults.**

| Pseudonym of relative | Relative's Age | Relationship of relative to autistic adult | Adult's Gender and age (years) | Age of Adult at time of diagnosis | Adult's autism spectrum diagnosis & LD | Co-occurring mental health reported | Adult's Education level attained |
|---|---|---|---|---|---|---|---|
| Mr A | 63 | Father | M, 22 | 21 | Autism & ID, developmental delay, dyslexia, dyscalculia, dysgraphia | Bipolar | No formal qualification |
| Ms C | 42 | Mother | M, 21 | 19 | ASD | | Higher education |
| Ms J | 51 | Mother | M, 18 | 16 | AS | Anxiety, OCD | High School GCSE |
| Mr S | 78 | Informally Adopted Father | M, 29 | 13 | AS | Addictions | High School GCSE |
| Ms E | 49 | Mother | M, 25 | 3 | Autism & ID | Anxiety, Epilepsy, OCD | No formal qualification |
| Ms P | 64 | Mother | F, 38 | 38 | ASD | Depression | Higher education |
| Ms R | 60 | Mother | M, 36 | 36 | ASD | Epilepsy | Basic School Qualification |
| Ms T | 49 | Mother | M, 21 | 5 | Atypical Autism & ID, developmental delay, dyslexia, dyscalculia, dysgraphia & dyspraxia | Anxiety | No formal qualification |
| Ms H | 65 | Mother | F, 33 | 31 | ASD & ID, developmental delay | OCD | No formal qualification |
| Ms D | 60 | Mother | M, 23 | 21 | Autism & dyspraxia | Anxiety | Higher education |
| Ms I | 61 | Grandmother | M, 20 | 5 | Autism & dyspraxia | | High School GCSE |
| Ms B | 46 | Mother | M, 21 | 7 | Autism & developmental delay, dyspraxia | Depression, Anxiety | Basic School Qualification |
| Ms L | 59 | Mother | M, 24 | 6 | AS & dyspraxia | Anxiety, ADHD | High School GCSE |
| Mr W | 81 | Father | M, 55 | 51 | AS | Mental health problems, diagnosis unknown | Higher Education |
| Ms Y | 31 | Female sibling | F, 33 | 22 | AS | Depression | Basic School Qualification |
| Ms N | 56 | Female spouse | M, 57 | 55 | AS, ID | Depression, Anxiety | Basic School Qualification |

**Abbreviations:** Asperger Syndrome (AS); Autism Spectrum Disorder (ASD); Obsessive Compulsive Disorder (OCD); Attention Deficit Hyperactivity Disorder (ADHD); Learning Disability (LD).

diagnosed in childhood (n = 6), in adolescence (n = 2), in early adulthood before they were 30 years old (n = 3), and in later adulthood between 30–55 years old (n = 5) with a range of autism diagnoses. Learning disability diagnoses were reported as: intellectual disability (n = 5); developmental delay (n = 4), dyslexia (n = 2), dyscalculia (n = 2), dysgraphia (n = 2) and dyspraxia (n = 5). Co-occurring mental health diagnoses of the autistic adults, if reported, were as follows: Depression (n = 4); anxiety (n = 7); epilepsy (n = 2); bipolar (n = 1); ADHD (n = 1); obsessive compulsive disorder (OCD) (n = 3); addictions (n = 1); and non-specific mental health problems (n = 1). The following levels of educational attainment had been achieved: higher education awards (n = 4); high school qualifications (all at GCSE level) (n = 4); basic school qualification (n = 4); or no formal school qualification (n = 4). Fifteen interviews took place in the relative's own home, and one on university premises.

## Analytical themes

The thematic analysis of the autistic adult interviews identified six main themes. The themes are summarised in Table 3 and reported below with illustrative quotes selected from interviews with autistic adults and relatives.

**Diagnosis as validating yet limiting.** Many adults, but not all, saw receiving an autism spectrum diagnosis as an end-point that would provide validation and an explanation for feeling throughout their lives that they were 'different' from others. Reactions to the diagnosis varied:

> I do a lot of research. I got so little information. I couldn't understand anything, but autism was the first diagnosis that actually sounded like that was what I'd been looking for all my life. (Ms E, 36, AS, AaD: 33)

> It was a bit traumatic at first. I didn't know what on earth was going to happen to me. I felt at the time as if my life was falling apart but I realised, "No, I'm just the same person I've always been." (Mr W, 61, AS; AaD:58)

**Table 3. Summary of qualitative themes.**

| Theme | Descriptive summary |
|---|---|
| Diagnosis as validating yet limiting | Providing validation and explanation<br>Engendering self-acceptance<br>Route (or not) to accessing support<br>Relief for relatives |
| Supportive and non-supportive social agents | Family relationships could be supportive and non-supportive<br>Impact of relationships in education<br>Employment: supportive and non-supportive<br>Employment: adjustments & 'being valued'<br>Health and social care: problematic access<br>Health and social care: mislabelling |
| The "invisibility" of the needs of autistic adults | Lack of understanding from others (including family)<br>Autism hidden by 'cleverness' & high functioning |
| Health in the context of autism | Co-occurring conditions: number & complexity |
| Staying 'outside' the circle | Just not fitting in<br>Stereotypical (mis)understandings<br>Learning to conform |
| Multiple lives with autism: imagined and lived | Lives 'wasted'<br>Missed opportunities<br>Conventional standards of 'success'<br>Satisfaction with life and achievements |

*I rejected the idea [of having autism] at first because I didn't think I could possibly get through to this stage in life and not know. (Ms K, 23, ASD, AaD: 22)*

Although individuals diagnosed in later life could find a diagnosis helpful in making sense of their experiences during their lives, even those diagnosed in early adulthood (such as Ms K) could take time to accept it. Many participants viewed the autism diagnosis positively, and expressed that it allowed them to be more accepting of themselves, for whom they are:

*"I'm different, but am I good different or bad different?" It [the diagnosis] kind of gives an answer that says "You're good different, it's nothing to do with being damaged goods. It's nothing to do with being faded or having character flaws that you can't escape from. . .but knowing that it's a neurological kind of thing is actually quite empowering, I think. (Ms H, 46, AS, AaD: 43)*

For some, the diagnosis was seen as a route towards receiving support or information for themselves and/or family; or information about rights and benefits. However, many participants reported either not receiving (adequate) information following diagnosis, or receiving information that was incorrect, out of date or not appropriate to their needs:

*I received a big pack. It had loads of out of date information about groups and people to contact. It was about this group and then I contacted them and they said they had nothing to do with it anymore but contact this person. I contacted that person. Then they said, "We have nothing to do with it anymore, contact this person." I was like, "I just won't bother." (Ms R, 39, ASD, AaD: 39)*

The example provided by Ms R reflected the frustrations of other interviewees, who found that the health and social care support they had expected after diagnosis was not forthcoming, or was unsuitable–for example, support groups scheduled during working hours, or for individuals with a different set of needs. A lack of support available for helping partners and family members to understand the individual's autism better was highlighted. Several participants described how the autism diagnosis changed their outlook on life *(see 'multiple lives')*.

Relatives who were parents of an autistic adult described how learning about their offspring's diagnosis often came as a relief, and vindication that the challenges faced by their son or daughter were not related to poor parenting skills. For example:

*It was a bit of relief, actually, that we'd been right in thinking that it wasn't down to the way we'd brought him up. You go through all these scenarios in your head—that, "Is it something we've done?" although we've got four and he was the eldest. "Have you done something wrong to make him the way he is and is it the way he's been brought up?" In some ways, it was a bit of a relief to know that, all along, there's been this underlying problem. (Ms R, 60, Mother of son, diagnosed with AS at 36)*

In summary, although responses to an autism diagnosis varied, the impact of a diagnosis in adulthood was significant both to autistic adults, and relatives, whether accessing a diagnosis occurred in early or later adulthood. Diagnosis, however, did not necessarily lead to expected and hoped for access to needed support.

**Supportive and non-supportive social agents.**   Relationships with family members were pivotal for participants, although experiences differed. For some participants, relationships with partners were described as challenging:

*I would imagine we've fallen out [wife's name] and I, millions of times—she was going to leave me the next day if I didn't change. Somehow we've managed to hang in there. (Mr M, 68, AS, AoD: 67)*

Whilst some participants described supportive relationships, others described how they had experienced their parents as '*not very nurturing*' or indeed '*as being quite brutal*':

*I've always known I was different, and my mam and dad didn't mind, anything I did they didn't mind, they were very accepting parents. I've got a photograph somewhere, I was three, of the toy gun because I wanted a toy gun. . . . . . . .That's sort of how it worked. (Ms A, 56, ASD, AaD: 42)*

*I think it's the fact that it's like people didn't pick up that made it worse. [. . .] I thought it was pretty much normal and I would have probably put all my problems down to my dad being just abusive (Mr S, 43, AS, AaD: 41)*

Experiences of relationships within the education setting had lasting impacts on individuals. Several adults and relatives shared their negative memories about lack of support whilst at school, and made reference to varying forms of 'labelling' by others in these contexts:

*In the end, it just all got too much and I refused to go to my lessons. Now I don't have any GCSEs or anything. One of the teachers there that was in charge of support, every time I get stressed I keep on remembering what she said to me when I was complaining about getting bullied because she went and said, "Stop complaining, nobody cares about you." (Mr U, 20, ASD, AaD: 6)*

*I had actually thought he [son] had Asperger's a long time before [diagnosis], but I was just a nutty parent, I guess, overprotective. Even though I'm highly educated, and I've worked in similar fields, I was ignored. I was made to feel that I was a failure as a parent, and I watched my son just deteriorate. Yes, so it was very traumatic. (Ms J, 51,mother of son diagnosed at 15 with AS)*

In the context of paid employment, individuals appreciated the help and support offered by some employers after diagnosis. However, some felt that declaring their autism diagnosis when applying for jobs may have limited their opportunities:

*I mean I would say that I did have a good job, which I got from quite early on and did until I was retired on health grounds. If in those days I'd had to put a handicap of any kind on the application form, it's quite possible that people could always find ways to get around things. I certainly think that that might have turned out differently. (Ms X, 65, AS, AaD: 63)*

Despite many employers being described by participants as supportive, adjustments to workplace practices considered 'the norm' were required to enable them to function well:

*I can't cope with the 10-hour shifts, they're too long for me. Also, if I do full-time—I was given the choice to move to full-time—but I realised I can't cope with the rolling shifts and the change from night to day, and earlies to lates, it just messes with my head too much, and I need too much sleep for that. So I'm doing five hour shifts, but I'm having to make up the money trying to do extra shifts, . . . . . . .So it's about finding your way, isn't it, really? (Ms D, 53, AS, AaD: 51)*

Despite the need for adjustments, some participants acknowledged that characteristics they ascribed to their autism–such as emotional detachment or an ability to maintain a focus on procedure–were valued by colleagues and could be an asset at work:

> *Where I work, people are very accepting. The senior management aren't, but the people I actually work with and for really embrace my illness, because what they say is that, regardless of what's happening, whether it's chaos or calm, my mood stays the same, it doesn't go up or down. It doesn't matter if they ask me to do something when it's 4:55pm on Friday. I'll respond exactly the same as if it's 9:10am on a Monday. (Ms N, 43, ASD, AaD: 41)*

Relationships with health and social service providers were often described as 'problematic'. A wide range of levels of available support and access to them were described. Participants also reported that gaining the support they needed often depended on identifying key individuals. Accounts included a range of experiences in accessing mainstream physical health, dental and mental health services, and social services, with some highlighting a lack of understanding/reasonable adjustments:

> *I have not had any support. I went to see my GP and he promised me a CPN (community psychiatric nurse). I have not heard anything. He promised me support and I have not heard anything. Yet my mum has had a stroke, but he is doing everything he can to help her. Yet he is doing nothing to help me. (Ms V, 54, ASD, AaD: 52)*

> *He [husband] has always been in employment and it is only market reasons why he hasn't been employed and he found that very difficult. There is no structure to his day. Then, when he signed on, it was a nightmare because they didn't understand at the dole office, they didn't even give him a special employment support allowance–it was Jobseekers. They put him through that rigmarole of looking for ten jobs a week, going on the computer, and he just panicked and he didn't understand the language of the computer. My son had to help and I had to help with job applications. That was extremely difficult but I just felt, after that initial help, you're out in the community again and it's not something you can tap into on a regular basis. (Ms N, 56, spouse of male diagnosed at 55 with AS).*

Overall, participants experienced both supportive and non-supportive relationships with family members, and in the employment setting. In relation to health and social services, experiences of delayed diagnosis, and lack of appropriate adjustments served to reinforce the feelings of both adults and relatives that autism was not a priority for service staff.

**The "invisibility" of the adult's autism.**   articipants described their life experiences with reference to the notion of the 'invisibility' of autism (relative to physical and visually apparent disability), and the ways in which this led to a lack of understanding from others.

> *To everybody else I've got two arms, two legs, and a head, and I'm normal. No I'm not. Not by a long shot. And it's that misunderstanding of people that makes my life absolute hell. (Mr J, 40, ASD, AaD: 48)*

> *I have spent most of my life feeling guilty that I haven't done all that I should have done or I have done wrong. I suppose there is a little bit of—not guilt but it is a bit sad that I didn't recognise that it could have been autism. (Ms H, 65, mother of daughter diagnosed at 31 with ASD)*

Our data demonstrate that relatives can express regret or, in Ms H's words 'guilt' or 'sadness', at not recognising their relative's autism. For the adults in this study, the extent to which

they experienced support or not, reflected a widespread lack of understanding–by family members and friends, schools, employers, health and social service providers–related to the invisibility of their condition and of their associated needs for support.

Family members and friends were included amongst those viewed as misunderstanding the individual's autistic condition because they couldn't 'see' it:

*Nobody believes me.* (*Husband's*) *parents don't accept it. My mum and her husband don't accept it. Then friends, people I work with tend to go "no, not you". I hear "they'll diagnose anyone nowadays" a lot because I can speak normally and I don't have any twitches. (Ms R, 39, ASD, AaD: 39)*

The experiences of lack of support during school and college were also attributed to this lack of visibility of their condition to others. A view that 'invisibility' of their condition could however, also stem from positively ascribed attributes such as 'cleverness' is supported by some participants' accounts:

*I remember back then just, you know, because nothing like autism, I think, was ever suspected because I was always really able with lessons and things. There were obviously students who were given a diagnosis and things, and I was never even looked at for anything like that because I was really clever. (Mr T, 27, AS, AaD: 25)*

Mr T's account–illustrative of the experiences of other participants–shows that, especially in an educational setting, an individual's autism may be rendered invisible by a prioritisation of academic ability over awareness of other challenges experienced by the individual. However, for those with a high level of academic functioning, this form of 'invisibility' can prevail throughout other aspects of life, affecting interactions with employment agencies, health and social care providers, employers and the like:

*They [agencies making decisions about support entitlements, e.g. benefits entitlement (DWP)] don't see the autistic when they're on their own in a room in tears, in sheer mental meltdown when they couldn't even put two and two together, when the brain is so stressed it can't do basic arithmetic or remember basic facts. The simplest things can then go out your head. It's that, it's the lack of awareness of how severe this condition can be, even if you've got a high enough IQ to be able to do complex calculations or learn multiple different languages, it doesn't matter. It's still really disabling (Mr S, 43, AS, AaD: 41)*

Mr S, above, makes a distinction between the private and public face of the individual with autism. During interview, Mr S also referred to the Department of Work and Pensions (DWP) as a *'major offender'* in seeing autism as *'not a significant disability'*. This conceptualisation of autism as invisible to professional providers was especially evident in interviewees' accounts of their encounters with health professionals and the health care service:

*. . .my GP kept fobbing me off with depression, depression, depression and that was it. (Ms Y, 24, ASD, AaD: 23)*

In summary, individuals–and relatives–spoke of different ways in which their autism was rendered invisible to friends, family members, school and college staff, employers, social service/welfare and healthcare providers. This invisibility was seen as a key reason for a lack of understanding by others, of their needs and for their behaviour.

**Health in the context of autism.** Most of participants described co-existing physical and mental health conditions and problems. Common mental health conditions included depression, anxiety, and obsessive-compulsive disorder. Like Ms B below, many had attempted to make sense of how these conditions related to each other, over the course of their lives.

*I'm very overweight; I have lots of indigestion problems. . . There's a lot of reading around this idea that autism is linked to things like indigestion and things like that, so I reckon that's probably linked, but it's also the stress is probably causing that, too. I don't go out, so I don't exercise a lot; I have migraines from stress. It's really interwoven: is it autism, is it depression? Well it's probably one caused by the other. (Ms B, 37, AS, AaD: 34)*

Ms P, below, reflected on the long term struggle her daughter (diagnosed in her late 30s) has had with mental health issues:

*She [daughter] has since she was 13 recurring depression bouts and anxiety. She has good times and bad times. But it is not diagnosed as anything but depression. It is not bipolar. I think now looking back on it now I think it is related to the fact that she is autistic and she has tried so hard all her life to change. (Ms P, 64, Mother of daughter, diagnosed with ASD at 38)*

Ms P, in her account above, reflects on whether her daughter's mental health problems have been associated with trying to address struggles that were related to her autism.

**Staying 'outside the circle'.** The negative impact of social communication difficulties and the consequences of misunderstanding from others, on participants' wellbeing and quality of life was frequently expressed. The experience of feeling 'outside the circle' was prevalent and pervasive:

*We had a school reunion last year and I went. It was good, I did enjoy it, but it was like there was a circle and I was out here sitting on the outside. For all people were speaking to me, I wasn't in that circle. I didn't fit in the circle, I just didn't fit. I was like an extra jigsaw piece that didn't go anywhere. (Ms L, 59, ASD, AaD: 57)*

This (self-assessed) judgement of 'not fitting in', stemmed from feeling unable to judge the intended meaning of communications from others, or the sense that they did not know how to communicate effectively with others.

*I will see other people, and some of my close friends saying things and getting away with it, and I just can't get away with saying it. I must say it differently, or wrong, or at the wrong time. (Mr I, 47, AS, AaD: 46)*

Adding to the challenges of communicating with others, participants again referred to ways in which autism–and the range of behavioural characteristics—are misunderstood. Even when an individual's status as a person with autism is known (to themselves or others), such misunderstandings persist and have impact:

*I think one of the biggest myths is that people with Asperger's don't care, they don't feel; they're robotic. If that were only the case. We care far too much; that's the problem–far too much. We take on the world's worries and that's part of the problem. (Ms B, 37, AS, AaD: 34)*

Autistic adults described responding to challenges of social acceptance and participation by working hard to conform to others' expectations, in social situations, on a day to day basis. 'Learning to conform' was a key strategy for many of our interviewees, developed over time:

> *"I learned how to do what people expected me to do. It was very hard and it was exhausting. Even now I don't really know which one is the real me and that is the scary bit. I think "flipping heck, is that one me, is that one me or is that one me? Take your pick."* (Ms L, 59, ASD, AaD: 57)

Ms L was one of several participants who described conforming in this way as 'exhausting', or 'energy-draining' *(Mr P, 49, AS, AaD: 47)*. Some participants gave examples of specific contexts, such as the workplace, where it could be especially challenging to be (or to stay) in the circle:

> *I mean, the bits I dislike most are just having to deal with people. Phone calls, I always get stressed before- I get stressed every morning, you know, every day there is just the stress of having to go to work. But whenever there's a phone call from another party, I lose a certain amount of time just being anxious about it. And then a certain amount of time afterwards recovering from the energy of having to put on a normal face and deal with people, sort of thing. So that takes quite a lot of time out doing that. (Mr P, 49, AS, AaD:47)*

Despite the positive responses from employers (such as the ability to focus as reported above) that some autistic adults experienced, engaging in work on a day to day basis remained difficult. Concerns about exclusion of individuals with autism from the work environment remain evident:

> *. . .whenever there's unemployment it's always convenient to exclude the people that don't fit in so it's no wonder that we're excluded. Then they'll actually be worried because they think they're excluding a lot of the creatives and a lot of the out of the box thinkers. . . but to people like us, there is no box. We couldn't think in the box if we tried. We can't find the damn box. (Mr S, 43, AS, AaD: 41)*

For individuals in our study, who were providing accounts of their lives across different life periods, feelings of not fitting in–or being 'outside the circle' were prevalent, and affected many aspects of their lives. Their responses to coping with the challenges of social engagement focused on 'trying to conform', which was described as exhausting and draining.

**Multiple lives with autism: Imagined and lived.**   Participants reflected on the lives they were living, as autistic adults. Several participants described their lives as 'a waste'; while others described failing to meet others' expectations about what they would achieve in life (e.g., because they were considered 'clever'), and yet others recounted having had a 'good' life despite challenges.

Participants reflected on how their lives (or that of their relatives) may have been different if they had received an earlier diagnosis of autism. Views on this were mixed, and not necessarily related to their age when diagnosed. Two participants (an adult, a relative) described a sense of missed opportunities:

> *If I'd been diagnosed in the early years, say from the age of about seven, maybe my life would have taken on a lot more meaning to me. Like I'm saying, alright I've lived two and a half*

*lives, but they've not been that meaningful. There's not a lot I can look back on and say, "That was great." (Mr X, 66, AS & ADHD, AaD: 64)*

*If he'd had some input when he was younger, when he first got the flat. . . Say, he'd had the diagnosis and he's still got the flat, from day one, somebody might have advised him about personal hygiene and about household skills, which he won't take from us. The issue that he's now got of hoarding things might never have ever arisen. That might have been a big help, but now, things have got so bad, I think it's going to take a lot to reverse. (Ms R, 60, Mother of Son, diagnosed at 36 with ASD)*

However, not all participants felt that earlier diagnosis would have made much difference to their lives. Some of the participants reflected on the positive aspects of their lives, reporting satisfaction with their achievements in different areas of functioning but often including consideration of aspects of their careers:

*I feel happy that I've kind of got a house for myself, I've got a relationship and I've got a job that I really enjoy, so I feel really lucky there. (Mr T, 27, AS, AaD: 25)*

*I'm quite fortunate in that I've got a good job, I've had a successful career, and I'm quite well-respected in the industry. I feel that I dropped into something which I was good at, and I was lucky to. . . If I think back, it could have so easily have gone a totally different way. (Mr I, 47, AS, AaD: 46)*

Other participants however reflected that 'living with autism', had in some ways limited their lives, leading to disappointments, or that they could not meet the expected conventions of success:

*Really I'd like a giant career and a car and loads of money and a massive social group and then set activities on set nights of the week. That's not going to happen for me*

*Int: Do you think that would make you happy?*

*It wouldn't make me happy but I would be ticking those boxes to be normal. (Ms R, 39, ASD, AaD: 39)*

Indeed, some adults expressed mourning for the life they had hoped to have. For some these 'imagined' lives related to apparent stereotypes of what a 'normal life' might be coupled with the notion that this was a standard against which they should judge their own achievements. However this view was not universal, and several participants reported satisfaction with their lives and achievements.

## Discussion

To date, we believe this to be the largest qualitative interview study with a diverse sample, reporting on everyday experiences of autistic adults of different ages from across the UK. Our study confirms and extends some previously reported findings from both UK and international studies concerning challenges associated with 'fitting in' [6,10,39]; but also offers insights about autistic adults' experiences related to the potential benefits and disadvantages of the 'invisibility' of autism and the lack of priority given to autism, perhaps especially in apparently able and articulate adults.

Our findings demonstrate that autistic adults report recognising *'difference'* between themselves and others [6,10,39], and of *'wearing the mask'* [6,9] or *'camouflaging'* [20,21] across all

stages of adulthood. In keeping with other studies that have explored the experience and impact of camouflaging in both men and women [20,21], men and women similarly described learning to conform to the social expectations of others, as a strategy for getting through life, and how exhausting this social communication work can be. Some women describe the 'huge effort' to assimilate and conceal their social difficulties to 'fit in' [40]. However men also experienced the challenges of, and exhaustion from, meeting the expectations of others across social settings from school/college to the workplace. Both men and women also talked about social pressures and negative experiences such as being bullied. Some reported feelings of not meeting the expectations of parents and family members. Lai et al. [20] have reported that in autistic men, the phenomenon of camouflaging was associated with more depressive symptoms. Reflecting on early experiences, women also reported the negative reactions of others in response to their struggles with academic activities. These findings reflect also those of Hull et al. [21] who undertook a focused study of experiences of camouflaging amongst 92 adults (55 females, 30 males and 7 'other') with autism spectrum conditions. In their study they developed a three-stage model of the camouflaging process consisting of *motivations* (to fit in and connect with others), the camouflaging itself (to include 'masking' and techniques of compensation) and *consequences* (that include exhaustion, challenging stereotypes, and threats to self-perception). Hull et al.'s model was developed through thematic analysis of responses to open-ended questions in a self-administered survey, that specifically targeted experiences of camouflaging. It is interesting to note that in our qualitative study using open questions about life experiences, camouflaging was a salient topic for autistic individuals in the context of everyday life. Further although we did not set out to directly explore participants' experiences in relation to their gender as in other studies [8], our findings indicate that the challenges to conform to the expectations of others are not gender-specific and do not appear to be age-related. We concur with Hickey et al. [10] that the life experiences of older autistic adults appear to have much in common with younger autistic adults. However, most of the adults in this sample had the experience of receiving their autism diagnosis in the relatively recent past. It may in future be possible to explore through the ASC-UK cohort whether and how attitudes and experiences change over time.

Recently published studies [9,10,22,23], have reported a range of reactions to receiving an autism diagnosis in adulthood, including fear, disbelief, disappointment, frustration, feelings of validation, and relief to individuals and their families. A recent qualitative study of older autistic adults (over 50 years of age) (n = 9 interviewees) that focused on the experiences and impacts of a recent diagnosis, reported similar experiences to those of our autistic adults–both in terms of benefits and limitations. As in other studies [23], our participants told stories of missed diagnosis in early years, misattribution of diagnosis to other (typically mental) health conditions, and accounts of the general lack of understanding from health service providers of both autism and additional mental health conditions, were provided. Whilst some participants had hoped that a diagnosis would explain some of the difficulties faced during their lives, others had hoped that a diagnosis would lead to support from services. As reported in other studies however [6,39], our participants expressed disappointment at the lack of (adequate) services available to them and their families. Griffith et al. [6] reported that all of their participants described their attempts at accessing formal support as unsuccessful, referring to a lack of knowledge and understanding of care professionals as an explanation for such difficulties. Our findings support this, and bring a new perspective that adults themselves (and the relatives) argue that not only is autism often apparently 'invisible' but that there is a lack of priority about autism amongst service providers for either facilitating a diagnostic assessment or considering how and whether autism might be relevant, when assessing and treating co-occurring problems.

Our study highlighted the variety of experiences autistic adults had in navigating both formal and informal relationships. For some, having warm supportive relationships with parents, siblings, and extended family reportedly improved their social skills and self-esteem. However, other participants described how their autism both affected and was affected by these early relationships; those with more difficult family relationships reported lacking in self-confidence, and seemed more likely to describe their attributes and behaviours negatively. Some relatives regretted the impact of not recognising autism, or misinterpreting behaviours, or felt they were criticised for being overprotective when they identified their relative's needs. Similar to other studies [22], many participants reported negative school experiences: difficulties with other pupils, high levels of bullying, and the impact of teachers who did not understand the child's needs. Our findings highlight that there is more to understand about how autism is experienced in adulthood, but it is also important to investigate how past negative experiences impact on relationships, functioning and how individuals respond to challenges in adult life.

As in other studies [6,25], autistic adults acknowledged that securing and maintaining paid employment was challenging, including those who reported previously 'doing well' academically. Some described the benefits of being in steady employment through their adult lives, but none reported that this was easy. The problems/challenges were often associated with the social interaction required at work. Several reported that they did not think they would have been offered their (current or previous) job if they had known and revealed their diagnosis at interview. Some described negative responses (including name-calling and bullying) from colleagues. However, once in their jobs, some of the adults expressed ways in which they were valued by their employers and employees (often for their 'autistic' skills) [41–43]. For the majority of those who disclosed their diagnosis at work, employers were described as putting strategies in place to support them. More inclusionary practices in the workplace are likely to assist autistic employees but may also benefit other employees, employers and organisations [44]. Our findings add support to those of a small, but recent, qualitative study by Soeker et al. that focused on experiences of autistic individuals transitioning from training into employment in South Africa [25]. Their study dentified a similar set of challenges experienced by autistic individuals whilst navigating what was described as 'the disorder' in the workplace, but as in our study, also reported positive practices, enablers, and successful adaptations.

Most of the autistic adults in our study reported experiencing distressing physical and mental health symptoms, consistent with other quantitative [45,46] and qualitative studies [10]. Depression and anxiety were the problems most commonly described in the interviews. A number of participants however, believed that they were inappropriately diagnosed with depression when they felt that their symptoms were due to their (previously undiagnosed) autism–findings that are now emerging from recent qualitative research with women diagnosed in later adulthood [23]. However, participants did not report experiencing awareness and knowledge about the likelihood of additional problems amongst service providers. More research is needed to understand the impact of co-occurring conditions, and improve how services address the needs of autistic adults with complex needs [18].

Our study provides new data concerning whether autistic adults consider themselves to have had a 'good' life. Whilst some participants did express a negative outlook, or made assessments of '*a life wasted*', others reported overall satisfaction with their lives and felt they had achieved in areas of their lives that included family and careers. A striking finding here was that such judgements tended to be made in relation to the expectations they felt that others held for them, or in reference to what might be considered 'normal' achievements in life, and not necessarily linked to their own aspirations. Our findings provide insightful reflections in relation to those of Cribb et al. [14] who provide evidence to argue that the normative standards for outcomes in life against which young autistic people are judged, must be reassessed.

Cribb et al. used semi-structured interviews to explore the experiences, expectations and hopes of young autistic persons (n = 26) and their parents (n = 28) around transitioning into adulthood. Whilst parents expressed concerns about the levels of support that their autistic offspring would continue to need across the lifecourse, young autistic adults themselves expressed much optimism about their developing futures. These young people described the process as providing opportunity for increasing independence and autonomy, and expressed a sentiment of *'feeling more in control of my life'*, whilst acknowledging that achieving these (self-identified) markers of a 'good life' would involve steps taken over time. These findings provide a useful contrast to those of our more diverse and older group of adults, many of whom were reflecting backwards rather than forwards on their lives, but similarly describe both challenges and achievements, and a need (for themselves and others) to recognise what are good and meaningful achievements in their lives.

A need for further research to focus on achieving practical improvements for enhancing the lives of autistic people, for example through interventions relating to life skills, vocational and educational opportunities, treatments and improved access to public services, has been documented [18]. Such improvements may address some of the reasons for the participants' negative assessments about their achievements in life that they often referred to as 'missed opportunities' in relation to school and education, and vocational achievements.

## Strengths and limitations

This study is likely to be the largest qualitative study undertaken with a purposive sample of diagnosed autistic adults, living in a range of UK geographical locations with differing local services. Relative to previous research [10,22] our study includes the accounts of participants from 18 to 71 years of age. Recruiting participants from a larger cohort study facilitated recruitment across a wide area (four regions of the UK), and allowed targeted sampling of interview participants across a wide age range. Further, including data from the perspective of relatives of autistic adults provided complementary information about the experiences of a further sample of autistic adults, of whom a small subsample have intellectual disability. However, despite this inclusive approach to study participation, the findings are likely to be most representative of participants in the average range of intellectual functioning diagnosed in adulthood, with capacity to undertake interviews (rather than of people with intellectual disability). A further strength is that the qualitative researcher, whilst covering the topics within the topic guide, provided the opportunity for participants to consider topics that were important to them. Participants were able to describe in detail the day to day challenges they experienced in communicating with others. Indeed as previously documented [4], participants wanted their voices to be heard as the 'experts on their own life'.

Limitations to the study also include the fact that the diagnostic status of our study participants was self-reported. However, other studies demonstrate a lack of difference in the experiences reported by participants with confirmed and non-confirmed autism diagnoses [10], or confirm the similarity of experiences of 'self-diagnosed' autistic adults, with those reported in this paper [47].

## Implications

The findings from this study contribute to a growing body of evidence demonstrating a clear need for improving the knowledge and expertise of service providers in relation to autism. Only once autism becomes positively visible and a priority topic for improvements in the quality, appropriateness, and access to healthcare and community resources including diagnostic assessment services for autism and co-occurring conditions alongside support interventions, is

it likely that a positive difference will be made to individuals' everyday lives, within families, in education, and in the workplace and community social settings. Our study demonstrated challenges, but also some positive examples of inclusion in activities that many autistic adults reported as challenging. These findings suggest that a shift of focus towards promoting citizenship, building on the skills and achievements of individuals, providing a personalised approach to 'reasonable adjustments' to the working and living environment, awareness of the social context and what people can contribute, facilitates engendering a sense of value rather than further stigmatising 'difference'. A participatory co-design approach to investigating how to improve service provision may help address some of the challenges participants report.

## Conclusions

This study is an important wake-up call for policy makers, commissioners of services and service providers both in the UK and internationally. In a recent international scoping review of diagnosis of autism in adulthood, the findings 'suggest that receiving a diagnosis in adulthood has a significant emotional impact, but accessibility and processes are inconsistent, and formal support services are lacking'. Further the authors highlight the need for more research on autism diagnosis in adults especially adult with intellectual disability [48]. In the UK autism specific legislation and clinical guidelines have been in place for the last decade to enable autistic adults to access the range of mainstream services they need. In the context of these findings, it is not surprising that the autism community remain concerned about the lack of service provision and service development [44]. New and existing initiatives focused on health care may address this somewhat, but will be limited in improving how autistic adults are supported in everyday life. Careful consideration is needed regarding what participation support can be delivered through social care, education vocational placement schemes, and voluntary sector support (including from other autistic people).

## Acknowledgments

We gratefully acknowledge contributions from our research participants, who generously gave their time to take part in the study; and from Jahnese Hamilton and Richard Hardy who supported data acquisition.

## Author Contributions

**Conceptualization:** Tracy L. Finch, Joan Mackintosh, Helen McConachie, Ann Le Couteur, Deborah Garland, Jeremy R. Parr.

**Data curation:** Joan Mackintosh, Alex Petrou.

**Formal analysis:** Tracy L. Finch, Joan Mackintosh, Alex Petrou, Helen McConachie, Ann Le Couteur, Jeremy R. Parr.

**Funding acquisition:** Tracy L. Finch, Helen McConachie, Ann Le Couteur, Deborah Garland, Jeremy R. Parr.

**Methodology:** Tracy L. Finch, Joan Mackintosh, Alex Petrou, Helen McConachie, Jeremy R. Parr.

**Project administration:** Joan Mackintosh, Alex Petrou, Helen McConachie, Jeremy R. Parr.

**Resources:** Jeremy R. Parr.

**Supervision:** Tracy L. Finch, Jeremy R. Parr.

**Validation:** Tracy L. Finch, Helen McConachie, Ann Le Couteur, Deborah Garland.

**Writing – original draft:** Tracy L. Finch, Joan Mackintosh, Helen McConachie, Ann Le Couteur, Jeremy R. Parr.

**Writing – review & editing:** Tracy L. Finch, Joan Mackintosh, Alex Petrou, Helen McConachie, Ann Le Couteur, Deborah Garland, Jeremy R. Parr.

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
