## [Decision Letter · Decision Letter 0]

24 Jun 2021

PONE-D-21-14051

“We couldn’t think in the box if we tried.  We can’t even find the damn box”: a qualitative study of the lived experiences of autistic adults and relatives of autistic adults

PLOS ONE

Dear Dr. Finch,

Thank you for submitting your manuscript to PLOS ONE. After careful consideration, we feel that it has merit but does not fully meet PLOS ONE’s publication criteria as it currently stands. Therefore, we invite you to submit a revised version of the manuscript that addresses the points raised during the review process.

Both reviewers believe that your paper has merit, and I find it very interesting too. The reviewers did, however, raise some important issues that need to be addressed. Please provide more detailed characteristics of the sample and respond to other comments made by the reviewers.

We look forward to receiving your revised manuscript.

Kind regards,

Ewa Pisula

Academic Editor

PLOS ONE

Journal Requirements:

Additional Editor Comments (if provided):

Reviewers' comments:

Reviewer's Responses to Questions

**Comments to the Author**

1. Is the manuscript technically sound, and do the data support the conclusions?

Reviewer #1: Partly

Reviewer #2: Yes

2. Has the statistical analysis been performed appropriately and rigorously? 

Reviewer #1: N/A

Reviewer #2: Yes

3. Have the authors made all data underlying the findings in their manuscript fully available?

Reviewer #1: No

Reviewer #2: Yes

4. Is the manuscript presented in an intelligible fashion and written in standard English?

Reviewer #1: Yes

Reviewer #2: Yes

5. Review Comments to the Author

Reviewer #1: The authors presented a large-scale, qualitative study to examine the life experiences of autistic adults living in the UK. Using thematic analysis, the authors came with six key themes reflecting autistic adults’ experiences in relationships, education, employment, access to services, sense of belonging, and general satisfaction with life. Autistic adults’ accounts were complemented with the analysis of the accounts provided by the relatives of autistic adults that were not able to directly participate in the study. The study is of high quality and covers an important though understudied area of autistic adults’ everyday life. However, some major issues must be addressed before the manuscript will be suitable for publication.

Major issues:

1) My main concern when reading the manuscript was related to the adequate representation of the autistic population that was studied. The authors aimed to include a diverse sample of individuals with a wide range of intellectual abilities and this was a rationale to include relatives of those adults who would be unable to express themselves in an interview. However, only six out of 16 relatives reported about individuals with learning disability. This raises a question, what were the reasons for no direct participation of the rest of this sample? Moreover, in my opinion, the experience of individuals with learning disability (or/and more severe autism symptoms) was not highlighted in the paper. This may lead to overgeneralization of the findings and suggest an overly unitary view of the autistic adults’ experience, which may not necessarily be true. Perhaps, relatives of autistic adults with learning disability spoke about experiences that were not in line with the key themes raised by individuals with no such disability (e.g., being dependent in self-care, living in a residential center, or receiving special education). According to the authors’ statement – these relatives’ accounts (not in line with the main themes) were purposively not included in the paper. In fact, autistic participants and relatives of autistic adults are not only two kinds of informants, but they report about two different populations. However, only the first population, of more able adults (i.e., direct participants), served as a basis for identifying the key themes. To sum up, I feel that the findings regarding people with learning disability or more severe autism symptoms (reported by the relatives) should be reflected more in the results, or this subsample should not be included at all to avoid the false notion of representing individuals who were not, in fact, fully represented.

2) The introduction sometimes lacked coherence and structure. For example, paragraph two starts with the topic of access to services which seems a bit out of place there and then moves to a different topic (lack of research about life experiences, starting from line 65) without a visible link between the two. The introduction should include a short review of existing literature (especially qualitative ones) related to the topic. Stating that this literature is scarce does not seem sufficient.

3) I lacked some important characteristics in the description of the participants, notably their education (commonly participants in this kind of research have higher educational attainment than the respective population). It would also be interesting to know if the participants had any co-occurring psychiatric or developmental conditions (the authors mention them later in the results but here more systematic data would be useful). Lastly, the authors included a geographical location in their purposive sampling so it would be good to report at least the size of the residential area where the participants lived (e.g., small/medium/big city, countryside). As the authors aim for the wide representation of the autistic adult population, I think this kind of information is important to provide.

4) In the beginning of theme 2, the authors write: “For some participants, relationships with partners were described as challenging” but there is no elaboration on that. It would be interesting to read more about romantic relationships as many of the participants lived with partners or spouses.

5) In the theme ‘Health in the context of autism’, the authors focus mainly on coping with depression and stress. If indeed a lot of participants had co-occurring neurodevelopmental disorders, it would be interesting to read about their related experiences. This might be a good place to include the experiences of people with learning disabilities (or their relatives). Then, in the discussion, the authors write that the most common co-occurring mental health problem was anxiety, which was not reflected in the results, so perhaps elaborating on this would be helpful as well.

6) The following part needs some correction: ‘Previously, ‘hiding difference’, was offered as a partial explanation for the difficulty identifying autism in girls and women. [citation needed] However there was little consideration of the burden on the individual nor that males and females might both struggle in this way [19].’ I think that the burden of camouflaging was recognized, including the study that the authors cite and some other studies. For example, camouflaging is known to correlate with depression in men (Lai et al., 2016). There is also at least one, large qualitative study on camouflaging in autistic adults that could be cited here (‘“Putting on My Best Normal”: Social Camouflaging in Adults with Autism Spectrum Conditions’ by Hull et al., 2017).

Minor issues:

1) Interview topic guides should be included in the appendices. Knowing what questions were asked is important to put the results in context.

2) The tables with participants' characteristics are rather lengthy, so I would shortly sum up the data in the main text as well (how many of the participants had each diagnosis, how many years passed since the diagnosis, their employment, and living status).

3) It would be good to present the main themes and their summary descriptions in a table.

4) GCSEs (line 287) – please provide an explanation for non-British readers.

5) In different sections of the paper, the authors write that the study included participants ‘from across the UK’ or ‘from across England’ – please clarify that.

Other points:

The underlying data (transcripts of the participants' accounts) were not made available, but I consider the authors' explanation sufficient.

Reviewer #2: This is very good qualitative study. It deserves to be read widely.

As a practitioner and researcher working with this population most of the material and comment provided sounds familiar. So a key question is how original it is.

I agree there are few such qualitative studies on adults. There are a number of single case autobiographies. There are also likely to be chapters in books on autism in adulthood that reflect similar discourses (e.g. chapters 2 and 4 in Brugha T The Psychiatry of Adult Autism and Asperger Syndrome, 2018, OUP.)

Are the following similar studies considered:

A descriptive, qualitative study of the challenges that individuals with Autism Spectrum Disorder experience when transitioning from skills training programs into the open labor market in Cape Town, South Africa.

Soeker MS.

Work. 2020;65(4):733-747. doi: 10.3233/WOR-203127.

PMID: 32310205

Communication and AAC in the lives of adults with autism: the stories of their older parents.

Hines M, Balandin S, Togher L.

Augment Altern Commun. 2011 Dec;27(4):256-66. doi: 10.3109/07434618.2011.587830.

PMID: 22136364

Exploratory Study of Childbearing Experiences of Women With Asperger Syndrome.

Gardner M, Suplee PD, Bloch J, Lecks K.

Nurs Womens Health. 2016 Feb-Mar;20(1):28-37. doi: 10.1016/j.nwh.2015.12.001. Epub 2016 Feb 12.

PMID: 26902438

and possibly:

'I definitely feel more in control of my life': The perspectives of young autistic people and their parents on emerging adulthood.

Cribb S, Kenny L, Pellicano E.

Autism. 2019 Oct;23(7):1765-1781. doi: 10.1177/1362361319830029. Epub 2019 Feb 28.

Minor details and queries.

The term Procedures should perhaps be Selection of participants or Participant sampling?

Around 1% of the UK population has an autism diagnosis [1, 2]. This is not a correct reading of these two sources. And besides it is the population prevalence (the sources cited are correct) and not the number with a diagnosis (of which some will be false) that matter.

For clarity it is more correct to claim that this is 'the largest qualitative study undertaken with a purposive sample of DIAGNOSED autistic adults

living in a range of UK geographical locations with differing local services'. (a study limitation is that it does not include undiagnosed adults who may be the majority group).

How many other countries have as in the UK set out to support reasonable adjustments for people with autism across multiple governmental responsibilities such as welfare, employment, housing, as well as education and health? (With reference to '... This study is an important wake-up call for policy makers, commissioners of services and service providers. In the UK autism specific legislation and clinical guidelines have been in

place for the last decade to enable autistic adults to access the range of mainstream services

they need. I...'

6. PLOS authors have the option to publish the peer review history of their article (what does this mean?). If published, this will include your full peer review and any attached files.

Reviewer #1: No

Reviewer #2: No

---

## [Author Response · Author response to Decision Letter 0]

26 Nov 2021

Given the extent of the comments and responses required, these are fully detailed in an uploaded document identifiable as responses to reviewers' comments.

---

## [Decision Letter · Decision Letter 1]

21 Feb 2022

“We couldn’t think in the box if we tried.  We can’t even find the damn box”: a qualitative study of the lived experiences of autistic adults and relatives of autistic adults

PONE-D-21-14051R1

Dear Dr. Finch,

We’re pleased to inform you that your manuscript has been judged scientifically suitable for publication and will be formally accepted for publication once it meets all outstanding technical requirements.  

Kind regards,

Ewa Pisula

Academic Editor

PLOS ONE

Additional Editor Comments (optional):

After reading carefully the reviewed paper and the reviewers' assessments, I recommend that you consider some of the linguistic changes proposed by the Reviewer #1. I agree with the Reviewer that this should be your decision, as the debate on the appropriate language in the autism spectrum field is still ongoing.

Reviewers' comments:

Reviewer's Responses to Questions

**Comments to the Author**

1. If the authors have adequately addressed your comments raised in a previous round of review and you feel that this manuscript is now acceptable for publication, you may indicate that here to bypass the “Comments to the Author” section, enter your conflict of interest statement in the “Confidential to Editor” section, and submit your "Accept" recommendation.

Reviewer #1: All comments have been addressed

Reviewer #2: All comments have been addressed

2. Is the manuscript technically sound, and do the data support the conclusions?

Reviewer #1: Yes

Reviewer #2: Yes

3. Has the statistical analysis been performed appropriately and rigorously? 

Reviewer #1: N/A

Reviewer #2: Yes

4. Have the authors made all data underlying the findings in their manuscript fully available?

Reviewer #1: No

Reviewer #2: No

5. Is the manuscript presented in an intelligible fashion and written in standard English?

Reviewer #1: Yes

Reviewer #2: Yes

6. Review Comments to the Author

Reviewer #1: I feel that the Authors have addressed all my concerns sufficiently. On a final note, I'd like to ask Authors to review the language used in the paper for inclusiveness. Some of the terms, for example, in the first sentence of the introduction, seem overly medical ('disorder', 'severe symtomps' etc.) in the context of the overall Authors' approach. In fact, in others places Authors put 'disorder' in quotation marks suggesting a social construction. Similarly, terms such as 'high functioning' are regarded obsolete and non-specific. It's better to refer to individuals intellectual abilities, communicative abilities, or level of independence directly, instead of labelling them as 'low' or 'high' functioning. However, as the debate about the appropriate language when writing about autistic people is ongoing, I leave these changes to the Authors' discretion and accept the manuscript without further revisions.

Reviewer #2: On data access. Qualitative data should not be shared as it is impossible to conceal the identity of participants. I am content that the data should not be shared. I could see no indication in the manuscript that data is made available.

I have only reviewed the authors responses to my advice. It is entirely satisfactory; I have no further comments.

There is another reviewer. I have not examined that part of the material.

I have no comments for the authors.

7. PLOS authors have the option to publish the peer review history of their article (what does this mean?). If published, this will include your full peer review and any attached files.

Reviewer #1: **Yes: **Mateusz Płatos

Reviewer #2: **Yes: **Traolach Brugha

---

## [Editor Report · Acceptance letter]

4 Mar 2022

PONE-D-21-14051R1 

“We couldn’t think in the box if we tried. We can’t even find the damn box”: a qualitative study of the lived experiences of autistic adults and relatives of autistic adults 

Dear Dr. Finch:

I'm pleased to inform you that your manuscript has been deemed suitable for publication in PLOS ONE. Congratulations! Your manuscript is now with our production department. 

Kind regards, 

on behalf of

Dr. Ewa Pisula 

Academic Editor

PLOS ONE